# Pelvic Floor Functionality and Outcomes in Oncologic Patients Treated with Pelvic Bone Resection

**DOI:** 10.3390/cancers17162629

**Published:** 2025-08-12

**Authors:** Edoardo Ipponi, Pier Luigi Ipponi, Fabrizia Gentili, Elena Bechini, Vittoria Bettarini, Paolo Domenico Parchi, Lorenzo Andreani

**Affiliations:** 1Department of Orthopedics and Trauma Surgery, University of Pisa, Via Paradisa 2, 56124 Pisa, Italy; 2Department of General Surgery, Villa Donatello Hospital, Viale Giacomo Matteotti 4, 50132 Florence, Italy

**Keywords:** custom made prosthesis, bone tumor, sarcoma, metastasis, surgical mesh, rehabilitation

## Abstract

While the impact of pelvic surgical resections and reconstructions on musculoskeletal functionality and complication rates has been widely described, less has been written about their effects on the nearby pelvic floor. Our study aimed to evaluate the consequences of surgical resections of pelvic bone tumors on the function of the pelvic floor muscles, as well as the digestive, urinary, and genital systems. Our patients had an average mild pelvic floor dysfunction (mean PFIQ-7 = 11.7). Patients with discontinuity of the pelvic ring had a significantly higher grade of pelvic prolapse (M-line) and worse PFIQ7 scores. Our outcomes suggest that pelvic bone resections may lead to a mild but significant dysfunction of the pelvic floor. Respect for the pelvic floor during resections, careful reconstruction, and eventual rehabilitation after pelvic resections in orthopedic oncology is recommended to minimize the risk of pelvic floor deficits.

## 1. Introduction

Almost 15% of all malignant bone tumors occur in the pelvic bones, making them one of the most common locations for malignancy in the musculoskeletal apparatus [1]. These neoplasms can be either primary bone sarcomas or soft tissue neoplasms that affect the pelvic bones by contiguity or secondary lesions that reach the pelvis through blood flow [1,2].

Similar to other anatomical sites, the clinical presentation of bone sarcomas in the pelvis may include pain, swelling, noticeable limping, and eventually pathological fractures due to the reduced bone resistance in the involved bone segment. However, compared to lesions that arise from the appendicular body, pelvic primary sarcomas are frequently associated with less evident symptoms, which leads to larger lesion sizes at the moment of diagnosis [1,2,3].

Due to the local aggressiveness of the tumors, their size, and the complex three-dimensional anatomy of the pelvis, the surgical treatment of pelvic cancers represents one of the main challenges in orthopedic oncology. When approaching a sarcoma, orthopedic surgeons must perform a wide resection of the involved body region, aiming for wide resection margins to eradicate the disease and preserve as much healthy tissue as possible to minimize the resulting tissue gap and maintain local functionality [4,5,6].

Before the dawn of limb-sparing surgery, hindquarter amputation represented the only treatment available for malignant bone tumors of the pelvic girdle. Since the 1970s, the introduction of chemotherapy and radiation therapy, the development of CT and MRI scans, advances in bioengineering, and better surgical techniques have allowed limb salvage for an ever-growing share of patients with localized pelvic bone malignancies. Limb-sparing surgery, which represents the treatment of choice for the majority of bone sarcomas today, implies preserving the treated extremity and replacing the damaged bone with prosthetic, biologic, or composite implants [7,8,9]. In recent years, the introduction of intraoperative navigation and tailored cutting jigs has increased surgeons’ precision during resection [10,11,12,13,14]. From a reconstructive point of view, innovative, complex implants, including modern, custom-made prostheses, enable surgeons to restore patients’ native anatomy better [4,5,6,7,15,16]. These advances, combined with improvements in physical therapy and postoperative rehabilitation protocols, are yielding increasingly better postoperative functional outcomes for the musculoskeletal system [17,18,19,20]. The performance of the treated lower limb has consistently been one of the surgeons’ primary goals, given its significant impact on patients’ activities of daily living [20,21,22]. Major intra-operative complications involving the musculoskeletal apparatus, as well as significant damage to nerves, vessels, and endopelvic organs, have also been largely considered in modern literature due to their potentially catastrophic effects on a patient’s local outcome and even global health [23,24,25].

Instead, literature still lacks evidence on the impact that pelvic resections in orthopedic oncology have on the pelvic floor. This muscular structure closes the bottom of the abdominal cavity. It provides structural support to the urinary tract, the colorectal system, and both the male and female genital apparatus, thereby influencing their position and functionality [26,27,28]. For their location within the pelvic girdle, pelvic floor muscles could be at least partially harmed during some massive pelvic resections. Such damage could cause dysfunctions to the pelvic organs that are anatomically linked to the muscular floor, with a negative impact on patients’ quality of life. Incontinence of the urinary tract or the bowel, or pain caused by their daily activities, could represent major issues, especially for fragile oncologic patients. Genital disorders might also negatively change patients’ mating behavior and, in some cases, impair their reproductive capabilities. Such biological damages could therefore have adverse effects also on the psychological and social point of view [26,27,28,29].

Our study aimed to look beyond the oncological outcome, the surgical complications, and the musculoskeletal functionality frequently evaluated by most papers in the literature after massive resections of pelvic bone tumors. In this paper, we assessed the impact of surgical resections of pelvic bone tumors on the performance of the pelvic floor, digestive, urinary, and genital systems.

## 2. Materials and Methods

This single-center, retrospective study was conducted by the ethical standards outlined in the 1964 Declaration of Helsinki and its subsequent amendments.

Our study involved a review of all patients with bone tumors affecting the pelvic bones who required pelvic bone resections at our institution between January 2017 and January 2024. Inclusion criteria included (1) a confirmed diagnosis of primary sarcoma, locally aggressive bone tumors, or bone metastases; (2) involvement of the innominate bone without disease extension to the sacrum; and (3) clinical and radiological follow-up lasting longer than 12 months. Exclusion criteria included (1) preoperatively diagnosed systemic or local diseases impairing the function of the bowel, urinary tract, or genital apparatus, irrespective of the bone tumor; (2) known intraoperative iatrogenic damage or postoperative evidence of damage to the endopelvic organs; (3) intraoperative sacrifice of nerve roots responsible for innervating the anorectal system, urinary tract, or genital apparatus; (4) bilateral pubic resections; (5) planned two-stage surgeries; (6) later occurrence of reconstructions’ failures, including aseptic loosenings, infections, or mechanical breakages; (7) local recurrences necessitating further surgical resections or hindquarter amputation.

Pre-operative CT scans and MRIs were used to evaluate the anatomy of each patient before bone resection and the subsequent reconstruction. Tumor localization and resection sites were classified according to the Enneking-Dunham classification [30]. Cases were also divided into two groups: Those with their pelvic ring integrity maintained or restored (Group A) and those with discontinuity of their pelvic ring after surgery (Group B). The performed reconstruction of bones, articulations, and soft tissues was reported for each case.

No reconstruction was performed for bone gaps in pelvic areas not directly involved in weight bearing. In cases that required bone sacrifice in weight-bearing areas, reconstructions were performed with massive allografts stabilized to the native bones with plates and screws (Figure 1), allograft-prosthetic composites, 3D-printed custom-made prostheses (Figure 2), or iliac-stability ice-cone prostheses. The reconstructive approach was chosen based on the tumor width and resection site, as well as the patient’s prognosis.

Postoperative follow-up consisted of serial office visits, clinical evaluations, pelvic X-rays, and CT scans of the thorax and abdomen. Within the first two years after surgery, patients underwent outpatient clinical assessment and imaging evaluations every three months. Between two and five years after surgery, cases were evaluated at 6-month intervals. Between five and ten years after surgery, annual evaluations were planned.

At each patient’s latest follow-up, the most recent CT scans were used on sagittal view to calculate the Hiatal line, the Muscular line, and the anorectal angle. The Hiatal line (H-line; drawn from the inferior margin of the pubic symphysis to the posterior aspect of the anorectal junction and representing the diameter of the levator hiatus) measures the width of the levator hiatus. A standard H line measured at rest is typically less than 6 cm. Eventual widening can weaken the pelvic floor’s ability to support organs and increase the chance of pelvic floor prolapse. The Muscular line (M-line; extends perpendicularly from the pubococcygeal line to the posterior aspect of the H line and represents the vertical descent of the levator hiatus) indicates the relaxation or descent of the levator hiatus, therefore directly portraying pelvic floor prolapse. In healthy patients, the M-line measures less than 2 cm. An M-line longer than 2 cm is suggestive of pelvic floor descent. The two lines are often used in conjunction to assess the overall anatomical health and function of the pelvic floor.

The anorectal angle is the angle drawn between the central axis of the rectum and the central axis of the rectum. The normal values range from 70 to 134 degrees. It is considered a factor in fecal continence and defecation, influenced by the puborectalis muscle.

In cases where monolateral pubic resections were performed (involving the pubis symphysis), the contralateral side was used as a landmark for the anterior end of the H-line (Figure 3).

At their latest follow-up, patients’ clinical status regarding the functionality of their pelvic floor, urinary tract, bowel and rectum, and genital system was assessed using the disability score of the Pelvic Floor Impact Questionnaire (PFIQ-7) [31].

The PFIQ-7 is a short, self-reported questionnaire designed to assess the impact of pelvic floor disorders on patients’ quality of life. It consists of three subscales evaluating the Urinary Impact Questionnaire (UIQ-7), the Colorectal-Anal Impact Questionnaire (CRAIQ-7), and the Genital and Pelvic Organ Prolapse Impact Questionnaire (POPIQ-7). Each subscale contains seven issues, scored on a 4-point scale ranging from 0 (no negative impact) to 3 (huge negative impact). The sum of all points represents the final PFIQ-7 score. The PFIQ-7 was preferred to other scoring systems that assessed the functionality of the single systems alone [32,33].

Statistical analysis was performed using Stata SE 13.1 (StataCorp LLC, College Station, TX, USA). Correlations between continuous variabilities were assessed using the Pearson correlation test. Dependence between categorical variables was evaluated using chi-square tests or Fisher’s exact tests, depending on the size of the variabilities. Means in independent samples were assessed using the *t*-test for cohorts with normal distributions and the Wilcoxon Mann–Whitney U test for non-parametric data. Statistical significance was set at 0.05 for all endpoints.

## 3. Results

In the period of our research, thirty-eight cases with malignant and locally aggressive bone tumors received pelvic bone resections in our institution. One of these patients was excluded due to a pre-operative diagnosis of Amyotrophic Lateral Sclerosis (ALS). Two cases had local recurrences that required further demolition interventions. Two cases were excluded due to known pre-operative chronic kidney failure (Stage 5 CKD). One case was excluded due to major perioperative complications involving the endopelvic organs. A bowel perforation with subsequent acute peritonitis was treated with emergency laparotomy, toilette, and colostomy. Two other cases were lost in follow-up.

The remaining thirty cases met our inclusion criteria and were enrolled in our study. There were 18 females and 12 males, with a mean age of 40 years (10–76).

Six cases suffered from metastatic carcinomas originating from the thyroid (4), the lung (1), and the kidney (1). One more case had a pelvic localization of plasmacytoma. The remaining twenty-three patients had primary pelvic tumors. Six were diagnosed with chondrosarcoma, five with osteosarcoma, and as many with Ewing sarcoma. Two cases suffered from leiomyosarcoma and one from myxofibrosarcoma. The remaining cases were diagnosed with solitary fibrous tumor (2), desmoplastic fibroma, and Epithelioid hemangioendothelioma.

A type I resection according to the Enneking-Dunham classification was performed in eight cases. Three cases involved type I–II resections, five type II, and three type II–III. In eight cases, the resection was confined to the anterior pelvis (type III). In the remaining three cases, the resection involved all three bony components of the innominate bone, requiring massive resections (type I–II–III).

The bone reconstructions performed in our cohort are summarized in Table 1.

After resection, the pelvic ring was intact or continuous in 12 cases (Group A), whereas it had been interrupted in the remaining 18 cases (Group B). In nine cases, synthetic surgical meshes (Dipromed surgical mesh; DIPROMED srl, San Mauro Torinese, Turin, Italy) were used to reconstruct massive soft tissue gaps involving the pelvic floor (4 cases), the abdominal wall (3 cases), or tendons of the pelvic girdle (2 cases).

The mean post-operative follow-up was 56.9 months (13–110). At their latest follow-up, our patients’ mean anorectal angle was 98° (69–152). The mean H line was 6.9 cm (4.1–9.5), and the mean M-line was 3.9 cm (2.2–7). The mean PFIQ-7 was 9.4 (0–40). A summary of our raw results is reported in Table 2.

No statistically significant correlation was found between patients’ diagnoses, resections, or reconstructive approaches and the anorectal angle or H-line values. The M-line of those who had pelvic ring interruptions was significantly higher compared to those who had their pelvic ring intact, according to a two-tailed T-student test (4.4 vs. 3.1) (*p* = 0.0070) (Figure 4).

The mean PFIQ-7 scores for different tumor types, resection areas, closed or open pelvic rings, and reconstructive approaches are reported in Table 3.

The mean post-operative PFIQ-7 scores of patients with an intact pelvic ring were significantly lower compared to those with an interrupted pelvic ring, as determined by a T-student test (3.4 vs. 13.4) (*p* = 0.0320) (Figure 5). In our cohort, no statistically significant difference in PFIQ-7 could be found between the singular different reconstructive approaches.

In our cohort, a Pearson correlation test also highlighted a statistically significant linear correlation between the M-line length and the PFIQ-7 score value (R = 0.902; *p* = 0.0001), suggesting that longer M-lines were associated with worse pelvic floor clinical functions (Figure 6). No other statistically significant correlation was detected in our cohort.

## 4. Discussion

Over the last decades, literature has bloomed with evidence about the clinical and functional impact of resections and reconstructions in patients with malignant pelvic bone tumors [1,5,6,9,10]. Since the dawn of limb-sparing surgery, surgeons have focused on the post-operative performance of treated limbs, assessed using a variety of functional scales; the most commonly used of these is the Musculoskeletal Tumor Society (MSTS) scoring system [34]. Innovations in implant designs, surgical techniques, and rehabilitation protocols have enabled progressive improvements in postoperative functionality [5,6]. Recent studies suggest that a large share of patients who underwent pelvic resections and reconstructions according to the latest reconstructive standards had good pelvic girdle and hip joint performances [15,35].

Intraoperative and postoperative complications represent another focus for orthopedic surgeons. Several authors reported on the most frequent complications in pelvic reconstructions. Infections represent a known issue in orthopedic oncology, particularly when using wide prosthetic implants. Loosening and mechanical complication rates vary depending on the chosen reconstructive approach. Soft tissue complications are another burden in the surgical treatment of pelvic bone tumors [23,24]. Preserving the main vessels and nerves, if not involved by the cancer, is a priority during both the resection and the reconstruction phase. The continuity of large pelvic arteries and their continuous blood flow is crucial for allowing perfusion of the pelvic girdle and the lower limb [36,37,38,39].

Furthermore, inadvertent vascular lesions in the endopelvic and gluteal regions may be challenging to stop; their massive bleeding could even lead to hemodynamic instability and put patients’ survival at risk [40]. When approaching the inner pelvis, orthopedic surgeons must also be careful to respect the endopelvic organs, including the colon and rectum, the bladder and the distal urinary tract, and the genital apparatus. Intraoperative or post-operative damages to the endopelvic organs are considered severe adverse events. Damage to these organs can impair their function, harm patients’ quality of life, and even put their survival at risk. In particular, leakage of urinary or fecal material in the pelvis may cause local infections that can easily evolve into localized acute peritonitis, which may progress to systemic peritonitis if left untreated [1,23,24,41,42]. However, the functionality of endopelvic organs and apparatuses does not depend only on their anatomical continuity, their vascularization, or their innervation.

The pelvic floor muscles span the bottom of the pelvis and support the pelvic organs, including the bladder, bowel, uterus (in females), and prostate and seminal vesicles (in males) [26,27,28]. The weakening of these muscles can result in a loss of structural support to these organs. The consequential organ dislocation and eventual prolapse may lead to urinary or fecal incontinence, genitourinary prolapse, sexual dysfunctions, and pelvic pain [26,27,28,29]. While the post-operative performances of the lower limb musculoskeletal apparatus have been largely investigated in modern literature, the impact of orthopedic pelvic surgery on the pelvic floor has long remained unnoticed.

Our results testified that major pelvic surgery in orthopedic oncology can imply partial, but still significant, alterations of the pelvic floor’s anatomical and functional status. Our mean PFIQ-7 score of 9.4 indicates a mild, yet negative, clinical impact on the pelvic floor in patients treated with massive pelvic bone resections. These outcomes, obtained in patients without significant preoperative disabilities or known intraoperative nerve or major pelvic organ damage, confirm that bone tumors and their treatment can compromise the anatomical continuity and functionality of the pubococcygeus, ileococcygeus, coccygeus, and puborectalis muscles. Locally aggressive tumors could spread from the bone to the nearby muscles. In other cases, when the tumor does not directly involve the muscle but comes close to it without a clear cleavage plane, surgeons might be forced to sacrifice the healthy tissue to achieve wide resection margins and increase the chances of disease eradication. In some cases, pelvic floor muscles, although intact, might be loosened due to the loss of bone insertion sites. Despite its following reconstruction, the iliac bone replacement might imply the detachment of the iliococcygeus muscle. Similarly, resecting the pubis, surgeons might release the anterior tendons of the pubococcygeus and puborectalis muscles. Furthermore, the indications for pubic bone reconstruction are not unanimous, and some authors prefer to avoid restoring the anatomical continuity of the anterior pelvic ring after resection [15,43,44]. Such an approach may potentially limit the anchorage sites required for subsequent tendon repair and healing. These conditions may be responsible for the longer M-lines and worse PFIQ-7 scores in our population, particularly in patients who experienced interruptions to their pelvic rings and subsequently lost some insertion sites.

We acknowledge that our study had some limitations. The rarity of pelvic bone tumors did not allow us to operate on broader populations, and the limited size of our cohort partially limited the statistical significance of some of the data associations we initially sought to investigate. Our cohort also had a high degree of heterogeneity, as different tumor types, pelvic segments involved, and reconstructions performed were gathered to allow an adequate sample size and faithfully portray the experience of our institution. The diversity in our cohort was also gender related, as our cohort collected both male and female patients, who have differences in their pelvic anatomies. An extension of our research to a multicentric scenario and more specific subgroups would be advisable in the near future.

The retrospective nature of our study represents another limitation, as it did not allow the complete standardization of the post-operative follow-up procedures for each patient. The patients in our study also had different follow-up times. Although the minimum follow-up time was set to 12 months after surgery, when patients’ clinical conditions should be stabilized, variations could be observed even more than a year after surgery. These limits should be overcome in the future with prospective studies to validate our outcomes further. Another issue might be represented by the absence of a dedicated scoring system for pelvic floor functionality after large surgical interventions, especially involving pelvic bones. The PFIQ-7 questionnaire, used in our study, is meant for and validated for degenerative pelvic prolapse. For this reason, the score is mainly designed to evaluate female patients, although the PFIQ-7 has already been successfully used to assess the pelvic floor functionality of male patients [45,46]. The development of specific scores would be desirable, as it could increase the quality of studies to come.

Beyond these limitations, our study provides an unprecedented focus regarding the impact of pelvic bone tumor surgical treatment on pelvic floor and lower abdominal wall functionality. Surgeons and patients together should be aware that the extensive demolitions required to treat primary and metastatic pelvic bone tumors may cause the onset of symptoms involving not only the musculoskeletal apparatus but also the urogenital and digestive tracts. Therefore, to prevent their onset and reduce their impact on patients’ quality of life, surgeons should not only focus on bone and joint reconstruction but also consider restoring the pelvic floor and abdominal wall to allow adequate post-operative functionality. Synthetic or biological surgical meshes could be used to fill soft tissue gaps and repair the wall’s tension [47,48,49]. Modern custom-made prostheses can also be designed with specific trabecular areas to facilitate muscle reattachment or dedicated holes to allow for direct suturing of the tendons [14,16,20]. Multidisciplinary treatment involving orthopedic surgeons and general surgeons trained in the repair of the abdominal wall and pelvic floor would be advisable, especially for the most complex cases.

Attention to the pelvic floor should also be extended beyond the surgical theater. In orthopedics and orthopedic oncology, physical therapy has a well-established role in enhancing and maximizing the functional outcomes of treated limbs and spines. Similarly, symptomatic patients could benefit from targeted pelvic floor muscle rehabilitation, which already represents a reliable treatment for other causes of incontinence.

## 5. Conclusions

The surgical treatment of malignant pelvic bone tumors implies the resection of large bone segments and eventually the sacrifice of a variable share of the nearby soft tissues. Beyond the movement organs, resections can also involve the pelvic floor, leading to a mild but significant risk of postoperative dysfunctions of the colorectal system, urinary tract, or genital apparatus, particularly in cases whose pelvic ring had been opened during surgical procedures. Respect for the pelvic floor during resections, careful reconstruction when necessary, and eventual rehabilitation after pelvic resections in orthopedic oncology are recommended to minimize the risk of pelvic floor deficits and increase the quality of life of complex and fragile patients.

## Figures and Tables

**Figure 1 cancers-17-02629-f001:**
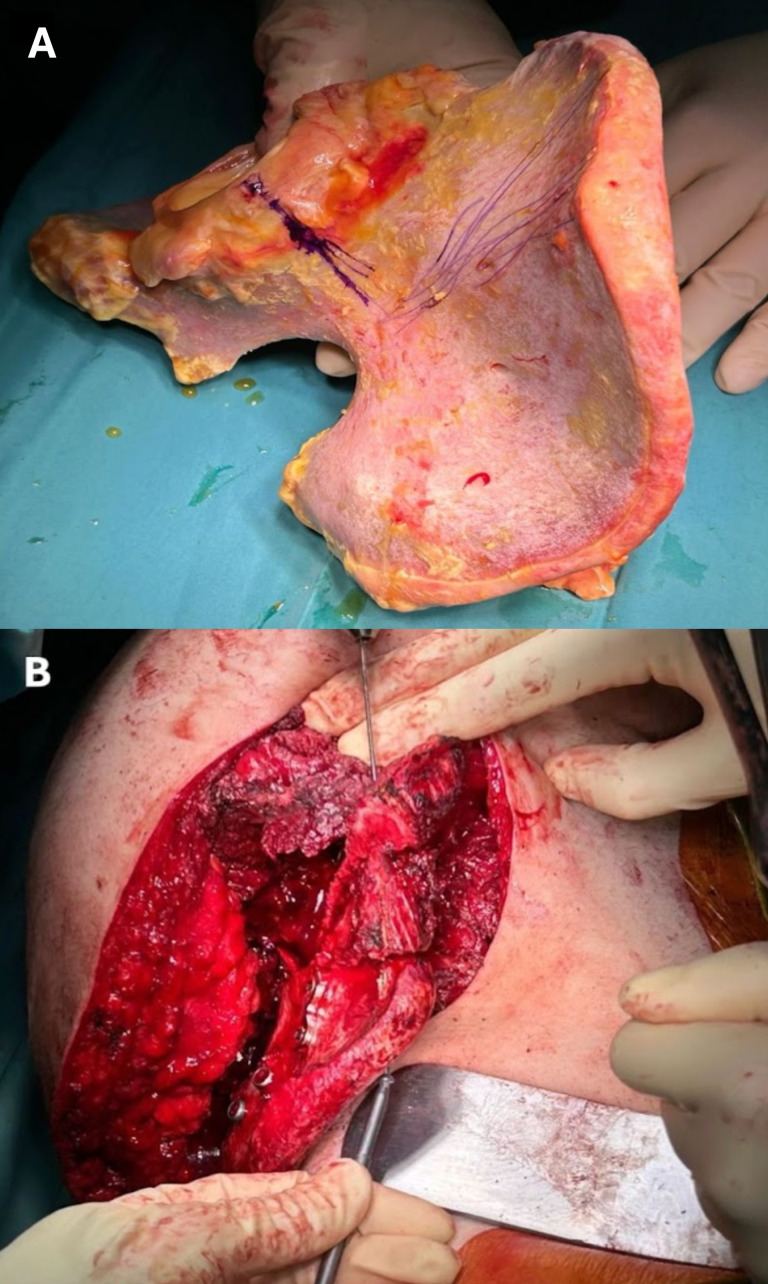
A massive pelvic bone allograft (**A**) shaped to replace the original supraacetabular and iliac regions of the innominate bone (**B**).

**Figure 2 cancers-17-02629-f002:**
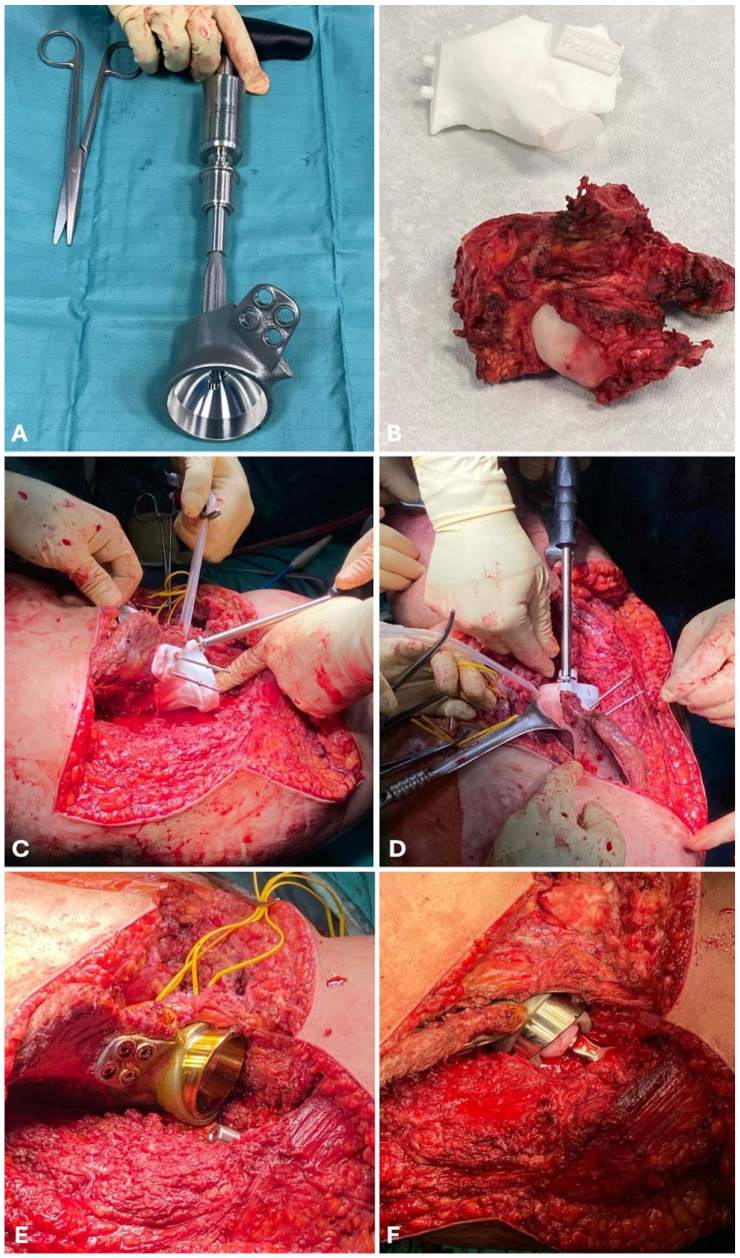
A 3D-printed custom-made prosthesis to reconstruct the periacetabular region (**A**). After an en-bloc resection of the bone tumor (**B**), a phantom implant with the same shape and size as the original bone is set to fill the bone gap (**C**,**D**). The phantom implant is temporarily stabilized to assess if the bone-implant interface is linear and steady. The phantom is then removed and replaced with the final prosthetic pelvic implant (**E**), and the prosthetic hip joint is finally stabilized (**F**).

**Figure 3 cancers-17-02629-f003:**
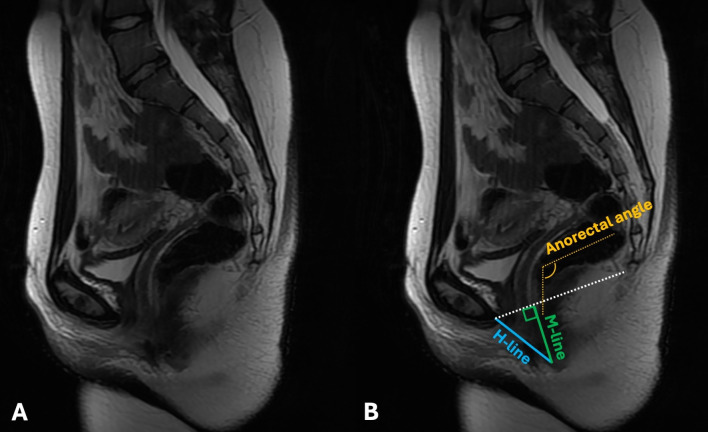
On the left, a sagittal MRI scan of a pelvis as it appears without notes (**A**). On the right, the same image with illustrations of patient’s anorectal angle (yellow), H-line (blue), and M-line (green) (**B**).

**Figure 4 cancers-17-02629-f004:**
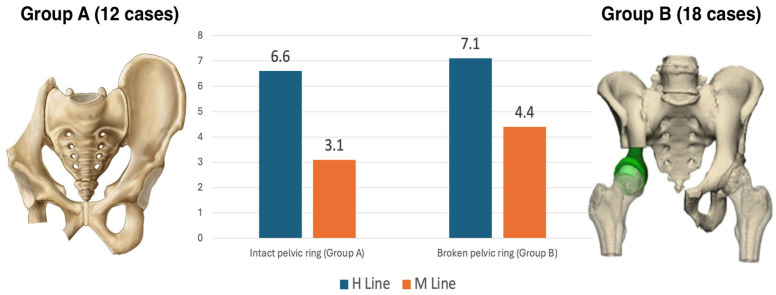
A graphic comparison of the mean H line (in blue) and M-line (in orange) values for those who had an intact pelvic ring (Group A; on the (**left**)) and those who had an interrupted pelvic ring (Group B; on the (**right**)).

**Figure 5 cancers-17-02629-f005:**
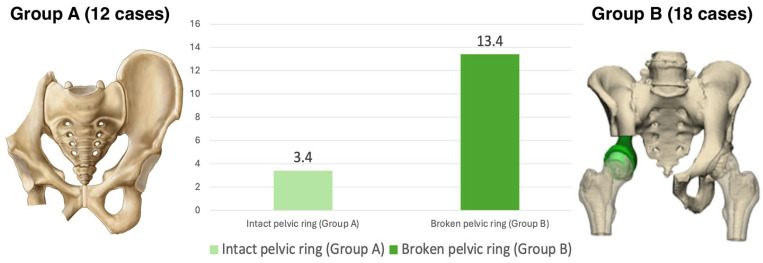
A graphic comparison of the PFIQ7 values of those who had an intact (Group A; light green) or interrupted (Group B; dark green) pelvic ring.

**Figure 6 cancers-17-02629-f006:**
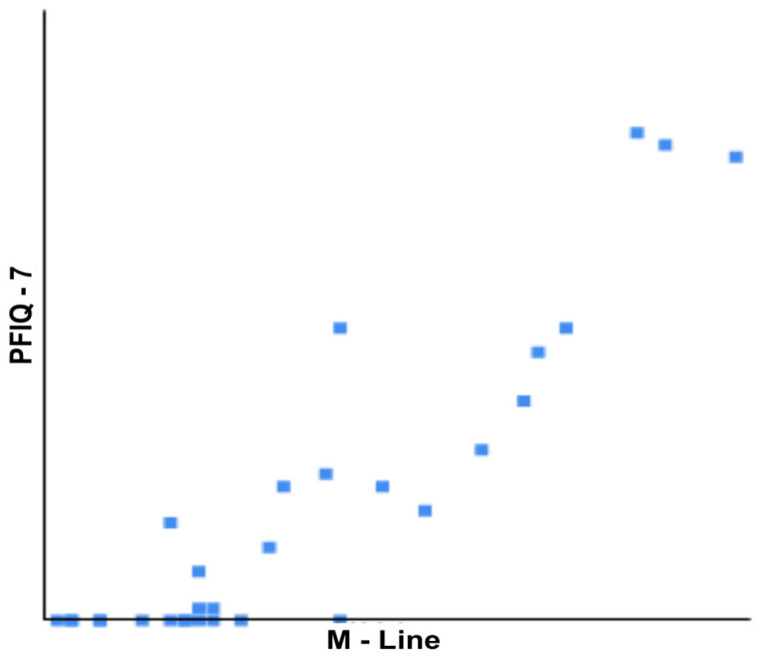
A graphic representation of the correlation between M-line values (*x*-axis) and PFIQ-7 (*y*-axis). A Pearson correlation score evidenced a statistically significant positive linear correlation between the two values in our cohort.

**Table 1 cancers-17-02629-t001:** A visual summary of all the reconstructive approaches for bone gaps in our cohort.

Reconstruction	Allograft	Allograft Prosthetic Composite	Custom Made Prosthesis	Ice-Cone Prosthesis	None
**N**	3	2	9	5	11

**Table 2 cancers-17-02629-t002:** A schematic resume of our cohort, divided into two groups depending on the intraoperative integrity or discontinuity of the pelvic ring.

	Total	Group A (Closed Pelvic Ring)	Group B (Open Pelvic Ring)
N	30	12	18
Age (Y)	40	42.8	38.1
Ano-rectal angle (°)	98	99	97
H-Line (cm)	6.9	6.6	7.1
M-Line (cm)	3.9	3.1	4.4
PFIQ-7 score	9.4	3.4	13.4
Follow-up (M)	56.9	68.6	49.0

(Y) = Years. (°) = Angle degrees. (cm) = Centimeters. (M) = Months.

**Table 3 cancers-17-02629-t003:** A schematic resume of our cohort’s final PFIQ-7 scores. Our patients were sorted for histological diagnosis, resection site (according to the Enneking-Dunham classification), pelvic ring closure, osteoarticular reconstruction performed, and use of surgical meshes.

	N	PFIQ-7 Score
**Tumor type (Diagnosis)**		
Ewing Sarcoma	5	0.2
Osteosarcoma	5	6.0
Chondrosarcoma	6	11.4
Metastases	6	17.7
Other	8	9.2
**Resection site (Enneking-Dunham classification)**		
I	8	3
I–II / II / II–III–	11	6.5
III	8	14
I–II–III	3	25
**Pelvic ring continuity**		
Closed (Group A)	12	3.4
Open (Group B)	18	13.4
**Osteoarticular reconstruction type**		
Custom-made prostheses	9	9.3
Ice cone prostheses	5	11.6
Allograft/APC	5	5.6
No reconstruction	11	10.2
**Pelvic floor reconstruction**		
Surgical mesh	9	2.7
No surgical mesh	21	12.3

N = Number of cases. APC = Allograft–prosthetic composites.

## Data Availability

The data that support the findings of this study are available from the corresponding author upon reasonable request.

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
