# Peer review of "Pelvic Floor Functionality and Outcomes in Oncologic Patients Treated with Pelvic Bone Resection"

_cancers, 2025, doi:10.3390/cancers17162629_

Round 1
Reviewer 1 Report
Comments and Suggestions for Authors
cancers-3758059
Title: The Impact of Pelvic Bone Resections and Reconstructions in Orthopedic Oncology on Pelvic Floors’ Functionality
General comments.
Many thanks to the authors for having presented their paper on pelvic floor functionality in oncologic bone pelvic surgery. The paper presents specific details related to the surgery with indications, clinical data and analysis of prognostic factors. The topic is interesting and useful for the orthopedic and oncologic community. However, there are critical aspects that should be considered. First of all, even if the Authors confirmed the limitations related to sample size and differences in terms of surgery and histotypes, it is really difficult answer to the aim of the study. I think that the comparison between a simple type I resection vs large acetabular resection is not adequate. The same regarding a solitary fibrous tumor vs a pelvic osteosarcoma (with chemotherapy, large soft tissue removal and reconstruction. Or again between a massive allograft vs an ice-cone prosthetic implant.
I would suggest Authors to focus their paper mainly on pelvic floor questionnaire giving more data and results on this topic. P.e. some tables or graphs on specific aspects of pelvic floor outcomes are needed and should be correlated to specific surgical details (bone reconstruction, pelvic ring closure, use of mesh and other).
Maybe the title should be “pelvic floor functionality and outcomes in oncologic patients treated with pelvic bone resection”. Otherwise, the paper does not reflect the expectancy of the title.
Moreover, consider the possibility of reducing the sample size on a specific subgroup limiting the bias of histotypes and surgical approaches.
Specific comments:
Check the correct period of evaluation. There is a difference between abstract and methods: We evaluated all malignant or locally aggressive pelvic bone tumors treated with bone resection 23 in our institution between 2017 and 2024… Our study involved a review of all patients with bone tumors affecting the pelvic 91 bones who required pelvic bone resections at our institution between 2017 and 2023
In the period of our research, thirty-eight cases with malignant bone tumors received pelvic bone resections in our institution… thirty cases met our inclusion criteria… But you included two solitary fibrous tumor and one desmoplastic fibroma (benign aggressive lesions).
Figures 4 and 5 do not reflect the sample size. Images are confounding
Sincerely, I would recommend that this paper be considered for publication after major revision. Greater clarity as to what they are actually reporting is needed. With this, the authors can provide a more direct discussion of the clinical implications of this work
Author Response
Dear Reviewer,
Thank you for your valuable assistance in enhancing the quality of our paper.
Please, find below the replies to your suggestions:
GENERAL COMMENTS:
“The topic is interesting and useful for the orthopedic and oncologic community. However, there are critical aspects that should be considered. First of all, even if the Authors confirmed the limitations related to sample size and differences in terms of surgery and histotypes, it is really difficult answer to the aim of the study. I think that the comparison between a simple type I resection vs large acetabular resection is not adequate. The same regarding a solitary fibrous tumor vs a pelvic osteosarcoma (with chemotherapy, large soft tissue removal and reconstruction. Or again between a massive allograft vs an ice-cone prosthetic implant.”
We agree that the heterogeneity of resections and reconstructions in our cohort represents one of the main issues in our paper. For transparency, this issue is now included in the limitations subsection within our discussion (LINES 336-342). The decision to consider and gather patients with different tumors, localized in different areas of the pelvic bone, and treated with different reconstructions was aimed at collecting a relatively large cohort, which could not be achieved otherwise due to the rarity of pelvic bone tumors. Furthermore, to our knowledge, the topic had not been previously covered in literature with case series; we prefer to include all possible cases, as our results could one day serve as a first reference for authors who, in the future, will provide larger information on the single tumor type, resection site, or reconstructive approach.
“I would suggest Authors to focus their paper mainly on pelvic floor questionnaire giving more data and results on this topic. P.e. some tables or graphs on specific aspects of pelvic floor outcomes are needed and should be correlated to specific surgical details (bone reconstruction, pelvic ring closure, use of mesh and other).”
We greatly appreciated your suggestion to incorporate new tables or graphs that highlight specific aspects of pelvic floor outcomes, correlated with specific surgical details. We introduced a new Table 3 in LINES 248-251 to increase the depth of our results.
“Maybe the title should be “pelvic floor functionality and outcomes in oncologic patients treated with pelvic bone resection”. Otherwise, the paper does not reflect the expectancy of the title.”
We changed our title, as suggested.
SPECIFIC COMMENTS:
“Check the correct period of evaluation. There is a difference between abstract and methods: We evaluated all malignant or locally aggressive pelvic bone tumors treated with bone resection 23 in our institution between 2017 and 2024… Our study involved a review of all patients with bone tumors affecting the pelvic 91 bones who required pelvic bone resections at our institution between 2017 and 2023.”
We included cases treated between January 2017 and January 2024 (included). We made the corrections in LINE 24 (abstract) and LINE 92 (Materials and methods). We apologize for the mistake.
“In the period of our research, thirty-eight cases with malignant bone tumors received pelvic bone resections in our institution… thirty cases met our inclusion criteria… But you included two solitary fibrous tumor and one desmoplastic fibroma (benign aggressive lesions).”
To fix this issue, we changed the phrase to a more correct “In the period of our research, thirty-eight cases with malignant and locally aggressive bone tumors received pelvic bone resections in our institution.” (LINES 192-193).
“Figures 4 and 5 do not reflect the sample size. Images are confounding”.
Figures 4 and 5 now reflect the sample size for both Group A and Group B.
Reviewer 2 Report
Comments and Suggestions for Authors
This article is interesting, generally well-written and properly documented. Some gaps should be addressed:
- The authors state that there is a lack of evidence in the literature regarding the impact of pelvic resections in orthopedic oncology on the pelvic floor; however, some relevant studies do exist and should be cited.
- The current introduction lacks focus on the specific topic of the paper - please revise
- The study includes intraoperative photographs and CT scans. Please clarify whether these images were fully anonymized and whether informed consent was obtained for their use
- Please confirm whether a formal waiver was issued by the Ethical Committee, in accordance with ICMJE guidelines and local regulations
- The description of measurements requires further detail. Please spell out and define the H-Line (Hiatal Line) and M-Line (Muscular Line), including their anatomical significance, the purpose of their measurement in this study, and the normal reference ranges
- Please provide a brief explanation or interpretation of the PFIQ-7 score
- The statistical analysis section should be expanded. Specify the statistical tests used, criteria for significance, and other relevant details
- The manuscript states that 38 cases were initially included, and after exclusions, 30 cases remained. However, the exclusions listed (1+2+2+1) total only 6, resulting in 32, not 30 cases. Please clarify this discrepancy.
- Line 153: The Enneking–Dunham classification should be cited appropriately.
- Paragraphs detailing the surgical protocols should be moved to the “Materials and Methods” ; also, consider summarizing surgical approaches per case in a table, as the current narrative format is difficult to follow
- Avoid repeating data that is already presented in tables; it makes the text harder to follow
- You did not address potential anatomical differences between male and female pelvises in your analysis of H-Line, M-Line, and their correlation with PFIQ-7 scores. While the sample size may be small, at minimum, this limitation should be acknowledged in the discussion
- Discussions should begin with a presentation and interpretation of your study’s results, followed by integration and comparison with existing literature. A more structured discussion linking your results to prior research would significantly strengthen the manuscript.
Respectfully submitted,
Author Response
Dear Reviewer,
Thank you for your valuable assistance in enhancing the quality of our paper.
Please, find below the replies to your suggestions:
“The authors state that there is a lack of evidence in the literature regarding the impact of pelvic resections in orthopedic oncology on the pelvic floor; however, some relevant studies do exist and should be cited”.
To our knowledge, this paper is the first to focus on the pelvic floor function in patients with pelvic bone tumors treated with pelvic resections. We could not find similar evidence in the literature on this topic. Please suggest references to us, and we will be pleased to examine them and eventually integrate them into our paper.
“The current introduction lacks focus on the specific topic of the paper - please revise”
We respect your point of view, but our introduction was not meant to immediately and only focus on the specific topic of pelvic floor functionality after pelvic resections in orthopedic oncology. Since ours represents a relatively new topic for modern literature, we used our introduction to provide context to readers, justifying our study and moving their attention from the “common paradigma” of orthopedic oncology, that mainly evaluates patients’ survival, musculoskeletal compliation and functionality, to the impact that the tumor and its treatment have on the pelvic floor, the urinary tract, the bowel and genital systems. Therefore, we consider our introduction to be conceptually correct. Following your suggestion, we have emphasized the adverse effects of pelvic organ dysfunctions and highlighted the aims of our study (LINES 85-93).
“The study includes intraoperative photographs and CT scans. Please clarify whether these images were fully anonymized and whether informed consent was obtained for their use”
Yes, informed consent was obtained, and images were anonymized. We have now made it clear in LINES 402-403.
“Please confirm whether a formal waiver was issued by the Ethical Committee, in accordance with ICMJE guidelines and local regulations”.
A formal waiver was issued in accordance with the ICMJE guidelines and local regulations. Material was provided to the Journal.
“The description of measurements requires further detail. Please spell out and define the H-Line (Hiatal Line) and M-Line (Muscular Line), including their anatomical significance, the purpose of their measurement in this study, and the normal reference ranges”
As suggested, we gave larger descriptions of M and H lines (LINES 148-162).
“Please provide a brief explanation or interpretation of the PFIQ-7 score”.
As suggested, we provided a brief explanation of the PFIQ-7 score in LINES 175-181.
“The statistical analysis section should be expanded. Specify the statistical tests used, criteria for significance, and other relevant details”.
As required, we expanded the statistical analysis section, specifying the tests used (LINES 184-188). Statistical significance cut-off had already been provided.
“The manuscript states that 38 cases were initially included, and after exclusions, 30 cases remained. However, the exclusions listed (1+2+2+1) total only 6, resulting in 32, not 30 cases. Please clarify this discrepancy.”
You are correct. Two patients lost in follow-up were missing. The error has been fixed (LINE 199). We sincerely apologize for the mistake.
“Line 153: The Enneking–Dunham classification should be cited appropriately.”
The article by Enneking and Dunham, published in 1970, is our new reference #30
“Paragraphs detailing the surgical protocols should be moved to the “Materials and Methods”; also, consider summarizing surgical approaches per case in a table, as the current narrative format is difficult to follow”
We moved details on surgical protocols to the materials and methods (LINES 123-139), as suggested. The number of cases that received a certain reconstruction is now reported in our new Table 1 (lines 215-216).
“Avoid repeating data that is already presented in tables; it makes the text harder to follow”.
We did not repeat the data reported in Table 2 (previously Table 1). The table is placed after the total data explanation (lines 222-225; lines 228-229), allowing for a more precise visual representation of the provided data. Furthermore, Table 2 provides not only the total data but also the separate data for Group A and Group B, offering readers more detailed information.
“You did not address potential anatomical differences between male and female pelvises in your analysis of H-Line, M-Line, and their correlation with PFIQ-7 scores. While the sample size may be small, at minimum, this limitation should be acknowledged in the discussion”.
We agree with you. We included the gender-related anatomical differences in the pelvis (LINES 339-342) among our limitations.
“Discussions should begin with a presentation and interpretation of your study’s results, followed by integration and comparison with existing literature. A more structured discussion linking your results to prior research would significantly strengthen the manuscript.”
As already mentioned above, to our knowledge, this paper is the first case series to focus on the pelvic floor function in patients with pelvic bone tumors treated with pelvic resections. We could not find previous case series in the literature on this topic. Therefore, we could not follow the structure you are suggesting.
If you could suggest sufficient evidence from previous studies reporting on the impact of orthopedic oncology surgery for pelvic bone tumors on pelvic floor functionality, we would be pleased to modify the structure of our discussion partially.
Reviewer 3 Report
Comments and Suggestions for Authors
This single-center retrospective study investigates an underexplored but clinically relevant topic: the functional impact of pelvic bone resections and reconstructions on the pelvic floor and associated organ systems. The authors highlight the risk of urinary, gastrointestinal, and genital dysfunctions following major pelvic surgeries in orthopedic oncology.
This is one of the few studies addressing pelvic floor functionality after orthopedic pelvic tumor surgery. It addresses a critical aspect of patient quality of life often overlooked in musculoskeletal oncology.
The study includes only 30 patients, which limits statistical power, particularly for subgroup analyses (e.g., type of reconstruction or use of meshes). A power calculation or post hoc analysis would help assess the reliability of the findings.
As acknowledged, the retrospective design introduces inherent biases in data collection, imaging intervals, and rehabilitation protocols. Prospective validation is needed.
The PFIQ-7 is validated primarily in women. While informative, its generalizability to men (40% of the cohort) is not addressed. This could impact the interpretation of the functional scores.
The cohort includes various reconstruction methods (custom prosthesis, ice-cone, allografts, no reconstruction), but the influence of each approach on pelvic floor outcomes is not explored.
The timing of PFIQ-7 and imaging assessments varies, with a wide follow-up range (13–110 months). Functional recovery may evolve significantly over time, which could affect comparability.
The manuscript could benefit from minor grammatical and syntactical corrections.
Author Response
Dear Reviewer,
Thank you for your valuable assistance in enhancing the quality of our paper.
“The study includes only 30 patients, which limits statistical power, particularly for subgroup analyses (e.g., type of reconstruction or use of meshes). A power calculation or post hoc analysis would help assess the reliability of the findings.”
“As acknowledged, the retrospective design introduces inherent biases in data collection, imaging intervals, and rehabilitation protocols. Prospective validation is needed.”
We agree that the size of our cohort, as well as the retrospective nature of our study, are both main issues in our paper. Our manuscript represents a first focus on the often-overlooked pelvic floor functionality in orthopedic oncology. From our point of view, power calculations should be considered soon, once other authors provide further evidence from different populations. Perspective validation will also be needed in the near future to develop our knowledge on this topic. We provided more emphasis on this point in the latest version of our paper (LINES 340-342, 348-349).
“The PFIQ-7 is validated primarily in women. While informative, its generalizability to men (40% of the cohort) is not addressed. This could impact the interpretation of the functional scores.”
We agree with your point. We added this aspect to consider among the limitations in our discussion section (LINES 349-355).
“The cohort includes various reconstruction methods (custom prosthesis, ice-cone, allografts, no reconstruction), but the influence of each approach on pelvic floor outcomes is not explored.”
We reported the mean PFIQ-7 of patients treated with different osteoarticular reconstructions in Table 1 (LINE 251). We also clarified in LINES 255-257 that, in our cohort, no statistically significant difference in PFIQ-7 could be found between the singular different reconstructive approaches.
“The timing of PFIQ-7 and imaging assessments varies, with a wide follow-up range (13–110 months). Functional recovery may evolve significantly over time, which could affect comparability.”
Although we had a minimum follow-up of 12 months to have reasonably stabilized clinical conditions, we agree with your point, as variations could be observed even more than a year after surgery. Therefore, we stressed this limitation in the latest version of our manuscript (LINES 345-349).
“The manuscript could benefit from minor grammatical and syntactical corrections.”
We corrected the syntactical and grammatical errors that we could find and we corrected them through the manuscript.
Please find the suggested corrections written in red in our revised manuscript. Grammatical corrections were written in blue.
Round 2
Reviewer 2 Report
Comments and Suggestions for Authors
The authors have performed satisfactory corrections and additions to the manuscript.
Reviewer 3 Report
Comments and Suggestions for Authors
The Authors made good efforts and the paper was significantly ameliorated.